

# MFEE: a multi-word lexical feature enhancement framework for Chinese geological hazard event extraction

Jie Gong,  Yang Cao,  Miao Zijing and  Qiaosen Chen

School of Computer Science, South China Normal University, Guangzhou, Guangdong, China

## ABSTRACT

Event Extraction (EE) is an essential and challenging task in information extraction. Most existing event extraction methods do not specifically target the Chinese geological hazards domain. This is due to the unique characteristics of the Chinese language and the lack of Chinese geological hazard datasets. To address these challenges, we propose a novel multi-word lexical feature enhancement framework (MFEE). It effectively implements Chinese event extraction in the geological hazard domain by introducing lexical information and the designed lexical feature weighting decision method. In addition, we construct a large-scale Chinese geological hazard dataset (CGHaz). Experimental results on this dataset and the ACE 2005 dataset demonstrate the approach's effectiveness. The datasets can be found at https://github.com/JieGong1130/MFEE-dataset. The code can be found at https://github.com/JieGong1130/MFEE-master.

## INTRODUCTION

In recent years, geological hazards have occurred frequently and seriously endangered human life and property safety. They have caused significant damage to the economy, resources, and environment. Event extraction can obtain valuable structured knowledge from the fragmented information on the Internet. It helps to enhance geological hazard warning and emergency decision-making. However, the current event extraction framework does not explicitly target the Chinese geological hazard domain. This is because of two significant challenges:

**Lack of data**: Most event extraction methods adopt a supervised learning paradigm. This paradigm relies on detailed human-labeled data, but there is no tagged corpus for event extraction in the Chinese geological hazard domain.

**Chinese event extraction**: The linguistic characteristics of the Chinese language make the task of event extraction more difficult. The reasons are as follows: (1) There is no natural segmentation between Chinese words. The different word segmentation should produce different semantics. (2) The trigger is a specific part of one word or contains multiple words, as shown in Fig. 1. It can easily lead to trigger-word mismatch problems.

In this article, we propose a multi-word lexical feature enhancement framework (MFEE) to address the challenges of Chinese event extraction. The key idea of MFEE is to introduce

Corresponding author
Yang Cao, yangcao@scnu.edu.cn

**(1)** 一名工人在地震中被 砸 死

**Type: Attack** **Type: Death**

A worker was killed in the earthquake.

**(2)** 房屋两侧出现 地面裂缝

**Type: Geofracture**

Ground cracks appear on both sides of the house.

**Figure 1** The trigger is a specific part of one word or contains multiple words.

lexical information into the character representation of the model using an external lexicon. Then, the importance of words in the lexicon is identified by the designed lexical feature weighting decision method. It can ensure that the lexical information that matches the current semantics is used reasonably. It also avoids the accumulation of errors in the model due to the wrong word segmentation boundary. To evaluate our proposed MFEE framework, we constructed a real-world Chinese geological hazard dataset CGHaz. Experiments conducted on this dataset and the ACE 2005 (*Doddington et al., 2004*) dataset demonstrate the effectiveness and generality of MFEE.

In summary, our contributions include:

(1) We propose a novel Chinese event extraction framework based on a character-level sequence labeling model but enhance it with word lexicon information.

(2) We constructed a large-scale real-world Chinese geological hazard dataset CGHaz, which can be used to address the challenge of lack of data in the field of Chinese geological hazards.

(3) We conducted experiments on the CGHaz and ACE 2005 datasets. The experimental results demonstrate the effectiveness and generality of MFEE.

## RELATED WORK

The current EE research can be roughly divided into two types: (1) joint execution of event trigger word extraction and argument extraction (*Li et al., 2014*; *Yang & Mitchell, 2016*; *Judea & Strube, 2016*; *Nguyen, Cho & Grishman, 2016*; *Liu et al., 2017*; *Sha et al., 2018*; *Nguyen & Nguyen, 2019*; *Zhang et al., 2019*; *Lin et al., 2020*). They solve the task by sequence tagging and extract triggers and parameters by tagging sentences only once. (2) Pipeline methods that first identify trigger words and then identify arguments at different stages (*Nguyen, Cho & Grishman, 2016*; *Li et al., 2020*; *Du & Cardie, 2020*; *Liu et*

*al., 2020*; *Chen et al., 2019*; *Carta et al., 2021*; *Li, Ji & Han, 2021*; *Huang et al., 2022*), but they usually lack a clear dependency relationship between trigger words and arguments. Also, they can be affected by error propagation. In recent years, leading trend research attempts to introduce semantic features of multiple dimensions to enhance the model's performance. *Chen et al. (2015)* applied convolutional neural networks, and *Nguyen, Cho & Grishman (2016)* applied recurrent neural networks to the event extraction task. *Zeng et al. (2016)* fused words' semantic information by combining a convolutional neural network and bidirectional long-term and short-term memory model. *Lin et al. (2018)* use dynamic multi-pooling convolutional networks to learn character and word representations separately. A gate mechanism is designed to fuse the two representations. *Zhu et al. (2020)* use words as input units to learn the characteristics of sentences and neighboring sentences using a network of two-way gated recurrent units. *Wu et al. (2021)* encode character and word representations separately to construct edges with their relationships. Fusing semantic information of characters and words using the graph convolution model. *Peng & Dredze (2016)* proposed a softword feature to enhance the character representation by embedding the corresponding segmentation label. However, most data sets do not have the golden section, and the segmentation result obtained by the divider may be incorrect. Therefore, this method will inevitably introduce segmentation errors. *Ma et al. (2019)* proposed a simple and effective method to incorporate word lexicon features without any complex sequence-structure design. Considering this problem, *Ding & Xiang (2021)* retain multiple-word segmentation results. However, if more than several segmentation results are discarded, it is easy to cause the loss of correct word semantics.

To address the above problems, our framework fully exploits word-level features based on the character-level sequence labeling model and retains all the matchable lexical information. With the word segmentation tool, we can ensure the accuracy of lexical information matching.

## Our approach

In this article, event extraction is formulated as a multi-class classification problem. Based on *Nguyen & Grishman (2015)*, entity types and corresponding event types or argument roles are assigned using the BIO annotation schema.

Our framework for extracting Chinese events consists of three phases: (i) the character representation module that encodes Chinese characters into the real value vector; (ii) the sequence modeling module that can effectively use both past and future input features thanks to a Bidirectional Long Short-Term Memory (BiLSTM) component; (iii) the extractor module, which extracts triggers and arguments using a conditional random field (CRF) layer. Figure 2 shows the architecture of our multi-word lexical feature enhancement framework.

## Character representation

The character representation module transforms the character into a real value vector. The output of this module consists of concatenating the following four features:

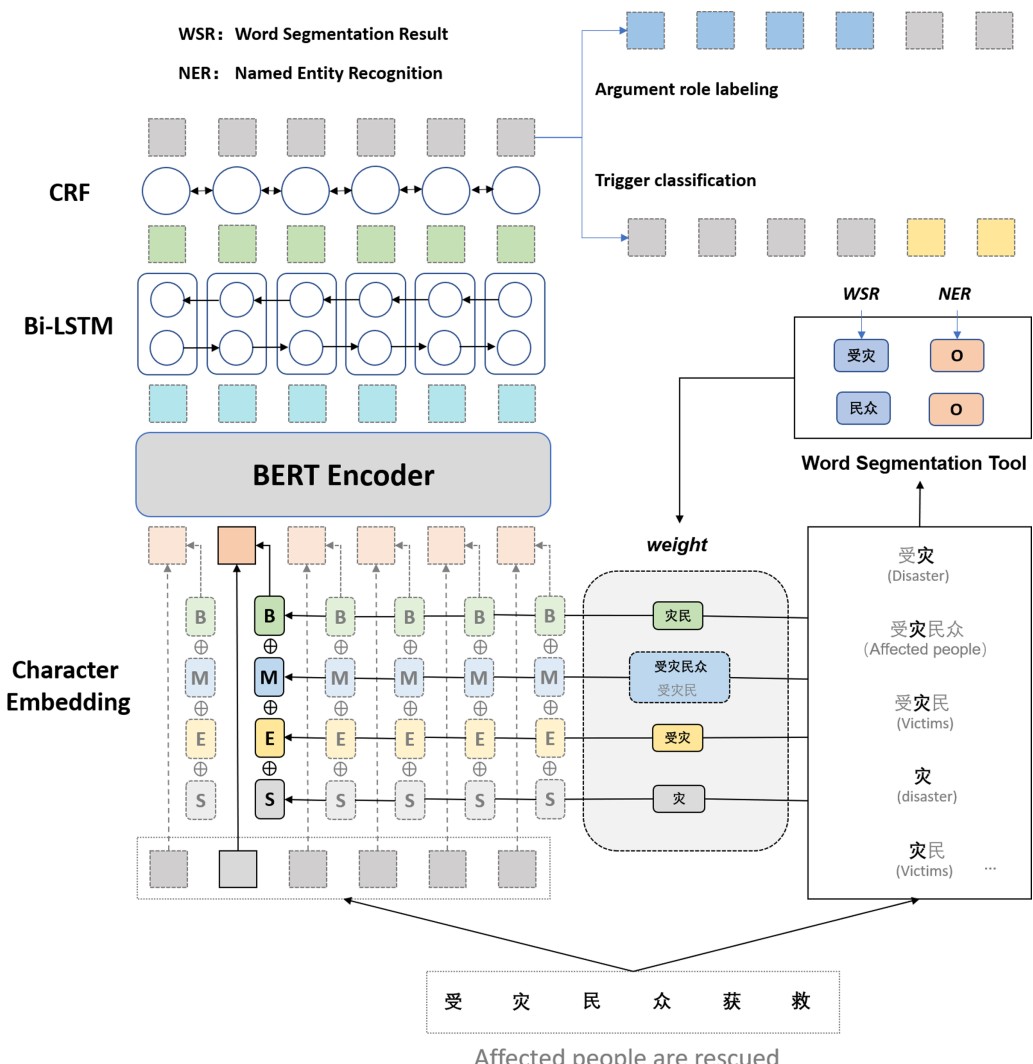

**Figure 2** The architecture of our multi-word lexical feature enhancement framework.

**Character vector features**

We transform characters into vector embeddings of distributed representations based on character-level modeling. BERT (*Devlin et al., 2018*) dynamically extracts character vector features based on context, making it more powerful for characterization. We use the BERT pre-trained model to obtain encoded representations of Chinese characters, which can better characterize the polysemy of characters. Given an input sequence $Input = c_1, c_2, \ldots, c_n$, where $c_i, i \in [1, n]$ is a character in the input sequence, then

$$x_i = BERT(c_i)$$

where $x_i$ denotes the vector representation of character $c_i$ obtained by the BERT model.

**Word segmentation features**

Frameworks based on word-level features alone cannot take advantage of the lexicon. It is easy to lose important local contextual information. Most of the previous work has

been done to obtain sequence label results by word segmentation tools to introduce lexical information. The corresponding labels are presented according to the relative position of the current word in the lexicon. Take the "BMES" tag as an example:

$$x_i = \left[ BERT\left(c_i\right); e^{seg}\left(seg\left(c_i\right)\right) \right]$$

where $seg\left(c_i\right)$ denotes the sub-label of character $c_i$ obtained by the automatic sub-labeling tool, and $e^{seg}$ denotes the lookup table of the sub-label "BMES".

**Multi-word lexical features**

Word segmentation tools can introduce lexical information related to the character context. However, if there is an error in the subsumption effect of automatic subsumption in the current context, it will misuse lexical information. *Ma et al. (2019)* introduce lexical features for the model by lexical matching in the named entity recognition task. Inspired by it, we use external lexicon matching to present rich multi-word lexical information for characters. This can avoid the accumulation of errors caused by a single lexical segmentation result.

For each character in the input sequence, we construct four sets, namely,{B},{M},{E},{S}, which represent the set of words in the corresponding position of the matched word. B records all words in the lexicon that can match the subsequence starting with the word. {M}, {E}, {S} denote the set of matching words where the word is in the middle of the word, at the end of the word, and a single character, respectively. Take Fig. 2 as an example; suppose the current subsequence fragment is "受灾民众获救" (Affect people are rescued); for the character "灾" in the lexicon, {B} matches "灾民" {M} matches "受灾民" and "受灾民众" {E} matches "受灾" and {S} matches "灾" If the set is empty, it is represented by a particular open word. The four groups are constructed as follows:

$$B(L) = w_{i,j}, \forall w_{i,j} \in D, 1 \le c = i < j \le n$$
$$M(L) = w_{i,j}, \forall w_{i,j} \in D, 1 \le i < c < j \le n$$
$$E(L) = w_{i,j}, \forall w_{i,j} \in D, 1 \le i < j = c \le n$$
$$S(L) = w_{i,j}, \forall w_{i,j} \in D, 1 \le c = i = j \le n$$

where D denotes all words in the lexicon that can be matched by the subsequence containing the character with subscript c.

Each word in the set is first encoded, and then the lexical in the set is represented in an aggregated manner. If a model dynamic weighting method is used to determine the word weights, it tends to increase the model complexity and the loss of time performance. Considering the temporal performance of the method, the frequency of occurrence in the lexicon is used as its weight. The lexicon can be the training set or the text of a vertical domain that belongs to the task requirements. The aggregated representation of each lexicon is then obtained as follows:

$$V^s(A) = \frac{4}{H} \sum_{w \in A} h_w e^w(w)$$

$$H = \sum_{w \in B \cup M \cup E \cup S} h_w$$

where $V^s(A)$ denotes the aggregated representation of word set A. $h_w$ denotes the weight of word w, that is, the frequency of occurrence in lexicon. $e^w(w)$ denotes the vector representation of finding word $w$ from the lexicon, while H denotes the sum of all word frequencies of the four-word sets. After the introduction of the multi-word lexical feature, the new character representation $x_i$ is

$$x_i = \left[ BERT(c_i); e^{seg}\left(seg(c_i)\right); e^l \right]$$
$$e^l = \left[ V^s(B); V^s(M); V^s(E); V^s(S) \right]$$

### Entity labels features

Entity tagging information can be introduced using word segmentation tools. The entity information can reflect the role that an entity fills in the sequence context. Considering that in most cases, the real data do not have labeled entity labels, we do not directly refer to the labeled entity labels in the dataset as in previous work. Instead, we introduce entity information based on the entity labels of the word segmentation tool. Take the Baidu LAC word segmentation tool as an example. The entity information can be labeled as "person name", "place name", "organization" and "time". The entity label information is introduced according to the entity where the character is located, and the final character is represented as:

$$x_i = \left[ BERT(c_i); e^{seg}\left(seg(c_i)\right); e^l; e^{entity}\left(entity(c_i)\right) \right]$$

where $entity(c_i)$ denotes getting the entity label corresponding to character $c_i$ and $e^{entity}$ denotes the lookup table of entity labels.

## Sequence modeling

The Long Short-term Memory (LSTM) model solves the gradient explosion and gradient disappearance problems of ordinary recurrent neural networks while training long sequences. It has good performance in capturing contextual sequence features. Bi-directional Long Short-term Memory (BiLSTM), as a combination of forwarding LSTM and reverse LSTM, can capture sequence semantic dependencies in both directions. Therefore, we obtain contextual features based on BiLSTM. BiLSTM splices the hidden states $\vec{h}_l$ and $\overleftarrow{h}_l$ calculated by the forward LSTM and reverses LSTM to get the remote state of character $c_i$:

$$\vec{h}_l = \vec{LSTM}\left( \vec{h}_{l-1}, x_i \right)$$
$$\overleftarrow{h}_l = \overleftarrow{LSTM}\left( \overleftarrow{h}_{l+1}, x_i \right)$$
$$h_i = \left[ \vec{h}_l, \overleftarrow{h}_l \right]$$

Input the obtained sequence representation $H = h_1, h_2, \ldots, h_n$ into the label sequence of prediction trigger words in the Label Inference Layer.

### Trigger and argument extractor

This module aims to predict subtypes of character nodes. Our trigger and argument extractor is proposed as a character-level classification task. We use Conditional random field (CRF) to predict labeling results, which takes into account the dependencies between consecutive sequence labels. The conditional random field adds constraints to the final predicted labels based on the information on the dependencies between the labels. These constraints are automatically learned through the CRF layer during the training process. Thus, based on the combined labeling constraints, the conditional random field selects the labeling results with high scores in the labeling list. The higher score indicates the labeling result is more credible, and the labeling with the highest credibility is selected as the final result. It is ensured that the prediction labeling sequence is reasonable and the probability of prediction error is reduced.

## EXPERIMENTS

In this section, we present empirical studies to answer the following questions: (1) what effect can MFEE achieve when facing the specific challenges of Chinese geological hazard event extraction? (2) What effect can MFEE achieve when facing the general challenges of Chinese event extraction? (3) How important are various components of MFEE?

### Experiment setup

**Data Collection with Event Labeling.** Considering the quality and authenticity of news texts, we use the news reports related to geological hazards at http://www.chinanews.com as the original corpus. It is used to perform geological hazard-based event labeling. We focus on seven event types: "earthquake", "collapse", "landslide", "debris flow", "land collapse", "land subsidence" and "ground fissure". They belong to the most common geological hazard events in the Classification and codes for natural disasters.

We consider the temporal dimension to classify the events in the field of geological hazards into five major categories. These are pre-hazard, hazard, Influence, Emergency Response, and Achievement events. This is further subdivided into sixteen specific types of events. The definition of events is shown in Table 1 (including trigger words and arguments).

Before the data label, we further preprocess the screened original corpus as follows:

(1) Sentence segmentation. We extract information based on the sentence level, such as labeling tasks and model training, so we need to cut the current news text into sentence form. We use " 。", "?", "!", ";", "......" etc. as separators to split the news text into sentences.

(2) Rewash the sentence. After the news text is segmented, many new sentences will be generated. The sentence text needs to be cleaned again, such as removing empty penalties, too long or too short sentences, repeated sentences, etc., to ensure the quality of the labeled corpus.

We use manual labeling to label the processed corpus. The labeling operation is simplified by the Jingdong Wise open text labeling tool. There are two stages to the labeling process. Firstly, the event trigger words are labeled, and then the arguments are labeled. We use

**Table 1  Definition of events on the CGHaz dataset.**

| | Event type | Description | Trigger words | Augments |
|---|---|---|---|---|
| Pre-Hazard | Reason 起因 | Reasons for the hazard | Swell, heat, storm 膨胀, 高温, 暴风雨 | Time, Place, Subject, Object |
| | Earthquake 地震 | | Earthquake, aftershock 地震, 余震 | |
| | Ground fissure 地裂缝 | | Ground fissure 地裂缝 | |
| | Landslide 滑坡 | | Landslide 滑坡 | |
| | Land subsidence 地面沉降 | | Land subsidence 地面沉降 | |
| Hazard | Collapse 崩塌 | Specific types of geological hazards | Collapse 崩塌 | Time, Place, Subject, Geologic hazard intensity |
| | Debris flow 泥石流 | | Debris flow 泥石流 | |
| | Land collapse 地面塌陷 | | Land collapse 地面塌陷 | |
| | Property damage 财产损失 | Property damage | Property damage 财产损失 | |
| | Damage 损失 | Buildings and infrastructure damage | Damage, destruct 损失, 破坏 | |
| Influence | Human life 伤亡 | Human life safety-related events | Injured, dead, disappear 受伤, 死亡, 失踪 | Time, Place, Subject |
| | Traffic 交通 | Events such as prohibition of passage or traffic congestion | Blocked, road interruption 阻塞,道路中断 | |
| | Movement 移动 | Movement of people's positions | Move, arrive, transfer, evacuate 前往, 到达, 转移, 撤离 | Time, Place, Subject, Departure place, Destination |
| Emergency Response | Investigate 调查 | Relevant personnel investigate the cause of the hazard | Investigate, understand, searched, troubleshoot, monitor 调查, 了解, 查找, 排查, 监测 | |
| | Action 措施 | Analysis of the hazard, relief and resettlement of the victims by relevant personnel | Resettle, establish, dispatch 安置, 成立, 调派 | Time, Place, Subject, Object, Object complement |
| Achievement | Achievement 成果 | Achievements of emergency response, such as people rescued, traffic restored, etc. | Rescue, restore 营救, 恢复 | |

a multi-person labeling method to increase efficiency and avoid incorrect labeling. We summarize some key parts of our labeling standard as follows:

(1) We expect different event types to be represented by different multivariate groups, and trigger words to be considered as unique identifiers of events.

(2) We do not only define events as actions or states that change in the real world. Some statements, such as policy notifications, are likewise considered events.

(3) We expect the fine-grained labeling of arguments to be complete, but not redundant. Determiners and modifiers for entities are retained only if they have a significant impact on the understanding of the event.

A total of seven people participated in the labeling process (including five labelers and two experts). For each document, we ask the labelers to label it independently. They can consult the experts if they have any doubts. After each batch of documents is labeled, the experts check them. Documents that do not meet the standards are returned for re-labeling. This process is repeated until the acceptance rate reaches 90%. The authors will check again after the expert check is completed. Documents that do not meet the standard are returned for relabeling until the acceptance rate reaches 95%.

Finally, we obtained 673 documents in total, and this number is larger than 633 of ACE 2005 dataset. There are 459 words on average in each document. We divided these documents into train, development, and test sets with the proportion of 8:1:1 based on the time order. Figure 3 shows the number of events of each type on the CGHaz dataset. Figure 4 shows the ratio for each geological hazard type on original documents.

**Evaluation Metric.** We use the same evaluation criteria as in the previous work. If the predicted trigger word has the same offset as the labeled trigger word, then the trigger word is correctly identified. In the case of the same offset, if the predicted trigger word is classified as the correct event type, the trigger word is proved to be correctly classified. We use Precision, Recall, and F1 values as evaluation metrics for the event extraction task.

**Hyper-parameter Setting.** For the input, we set the maximum number of sentences and the maximum sentence length as 32 and 128, respectively. We employ the Adam (*Kingma & Ba, 2014*) optimizer with the learning rate of 0.0015, train for at most 100 epochs and pick the best epoch by the validation score on the development set. We use large-scale Chinese with training word vectors as word vector lookup tables for word segmentation results. And it is used as a lexicon to obtain multi-word feature representation. The Baidu LAC tool obtains word segmentation results and common entity label information.

## Performance comparisons

**Baselines.** We compare the proposed Chinese event extraction model MFEE based on Multi-Word Lexical Feature Enhancement with the following event extraction methods:

(1) DMCNN (*Chen et al., 2015*): Feature extraction based on CNN and retaining the essential information of current trigger words and arguments through the dynamic multi-pool method.

(2) C-BiLSTM (*Zeng et al., 2016*): Fuse the semantic information of words by combining a convolutional neural network and BiLSTM model.

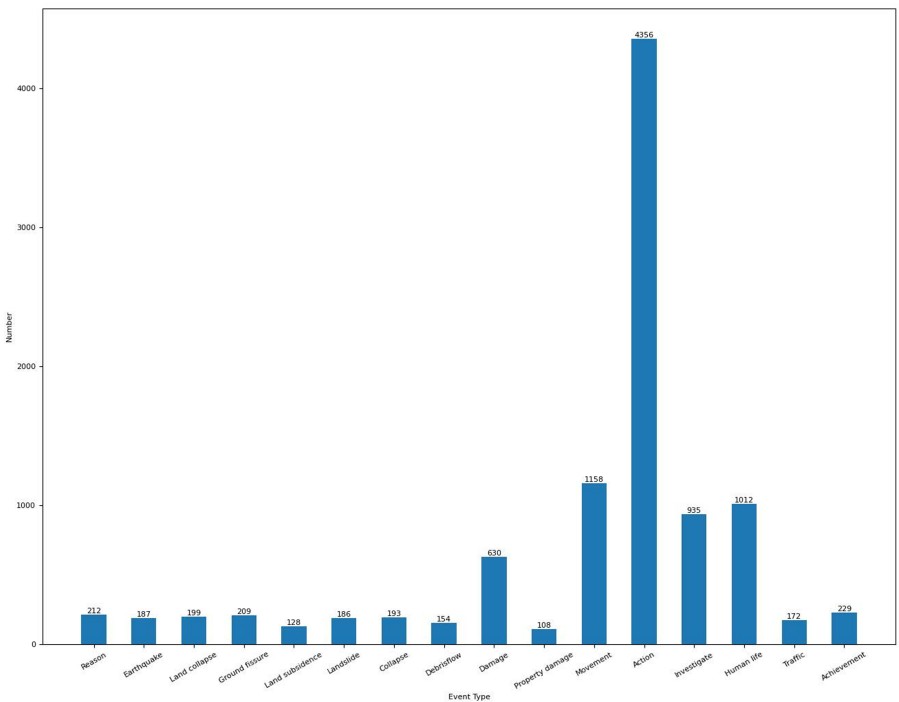

**Figure 3** The number of events of each type on the CGHaz dataset.

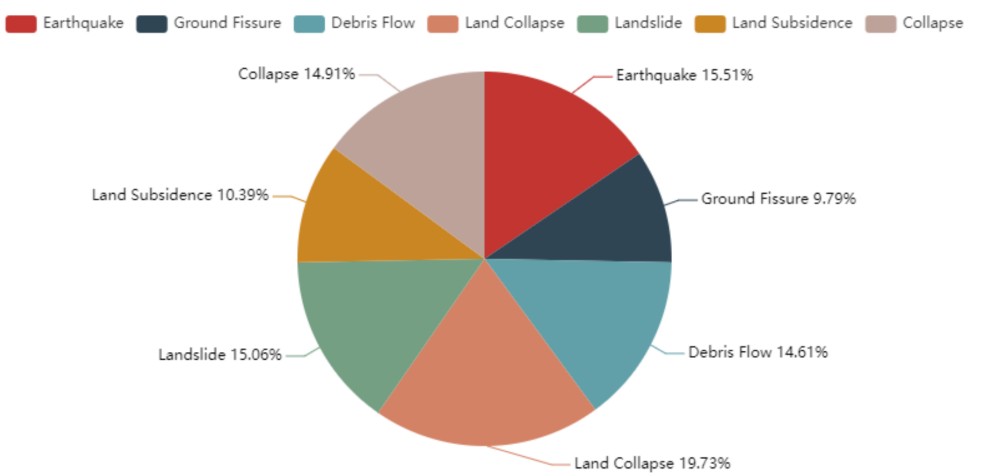

**Figure 4** The ratio for each geological hazard type on original documents.

(3)  BERT (*Devlin et al., 2018*): An event extraction model based on Bert's pre-training model connecting task classifier.

(4)  BERT+BiLSTM+CRF: Bert model extracts the word vector, obtains the sequence context semantic features through BiLSTM, and CRF receives the final event trigger word and argument prediction results.

**Table 2  Experimental results of event detection on the CGHaz dataset.** We conducted experiments on trigger word identification (TI) and trigger word classification (TC). Our results are reported with Precision, Recall, and F1 scores.

| | TI(%) | | | TC(%) | | |
|---|---|---|---|---|---|---|
| | Precision | Recall | F1 | Precision | Recall | F1 |
| DMCNN(word) | 73.7 | 71.6 | 72.7 | 68.9 | 66.3 | 67.6 |
| BERT-BILSTM-CRF | 88.4 | 84.0 | 86.2 | 86.2 | 82.6 | 84.4 |
| BERT | 85.6 | 85.3 | 85.5 | 83.1 | 82.8 | 83.0 |
| NPNS | 73.5 | 79.3 | 76.3 | 70.9 | 76.5 | 73.6 |
| C-BILSTM(word) | 77.7 | 73.0 | 75.2 | 74.6 | 70.2 | 72.3 |
| MFEE | 87.9 | 86.5 | 87.2 | 86.4 | 84.9 | 85.6 |
| MFEE(w/o segment) | 88.1 | 86.2 | 87.1 | 86.5 | 84.5 | 85.4 |
| MFEE(w/o BERT) | 87.2 | 84.9 | 86.0 | 85.8 | 83.4 | 84.5 |

(5) NPNs (*Lin et al., 2018*): Learning and fusing word and phrase representations using DMCNN. By triggering word generation block and event classification block in concert to complete event detection.

**Main Results.** As Tables 2 and 3 show, MFEE achieves significant improvements over all baselines on the CGHaz dataset. Figure 5 and Table 4 show the results of argument extraction for the seven geological hazard event types on the CGHaz dataset. Specifically, (1) MFEE improves 1.2 and 1.0 F1 scores overall best baseline BERT-BiLSTM-CRF in event detection and arguments classification, respectively. These improvements are mainly attributed to MFEE's approach to fusing multidimensional semantic features. It enhances the word vector representation. (2) DMCNN and C- BiLSTM were experimented with based on characters and words. The results show that the word-based model outperforms the character-based model on the event detection task. This reflects the importance of lexical features. (3) Compared with DMCNN and C-BiLSTM, the performance of the BERT-based character-level model is significantly improved. The reason is that the powerful characterization ability of BERT is fully utilized. It also avoids the problem of mismatches between words and triggers words. (4) Introducing BiLSTM for the BERT model to extract sequence contextual semantic features. And use CRF to constrain the output label can improve the performance of event extraction. (5) Although both NPNs and the model proposed by Ding Ling combine character and word information, NPNs only use lexical information from a single word segmentation result. It is susceptible to suffering from word segmentation errors. Ding-Ling's proposed model uses discard processing when the number of matchable words exceeds 3, which is prone to losing correct lexical information. Our method considers incorporating all matching words. The importance of lexical features is decided collaboratively by a designed lexical feature weighting decision method and word segmentation tools. In this way, accurate semantic information can be obtained.

As shown in Table 5, to verify the generality of MFEE, we evaluate the MFEE framework on the ACE 2005 dataset that contains 633 Chinese documents. We follow the same setup as *Zeng et al. (2016)*, in which 549/20/64 documents are used for training/development/test

**Table 3** **Experimental results of argument extraction on the CGHaz dataset.** We conducted experiments on argument identification (AI) and argument classification (AC).

|  | AI(%) | | | AC(%) | | |
|---|---|---|---|---|---|---|
|  | **Precision** | **Recall** | **F1** | **Precision** | **Recall** | **F1** |
| DMCNN(word) | 66.8 | 62.2 | 64.5 | 61.8 | 57.3 | 59.5 |
| BERT-BILSTM-CRF | 80.2 | 77.3 | 78.8 | 78.2 | 75.7 | 76.9 |
| BERT | 77.1 | 78.8 | 77.9 | 74.8 | 76.3 | 75.6 |
| C-BILSTM(word) | 70.1 | 66.7 | 68.4 | 66.8 | 62.3 | 64.5 |
| MFEE | 79.2 | 80.1 | 79.7 | 77.4 | 78.3 | 77.9 |
| MFEE(w/o segment) | 79.3 | 79.6 | 79.5 | 77.2 | 77.9 | 77.6 |
| MFEE(w/o BERT) | 78.5 | 78.6 | 78.5 | 76.8 | 76.7 | 76.8 |

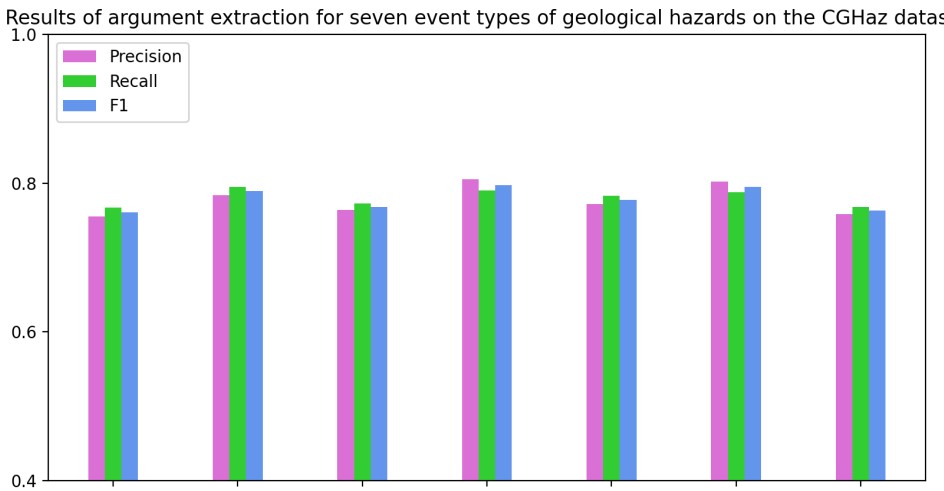

Results of argument extraction for seven event types of geological hazards on the CGHaz dataset

**Figure 5** **The results of argument extraction for the seven geological hazard event types on the CGHaz dataset.**

**Table 4** **Experimental results of argument extraction for seven types of geological hazard events on the CGHaz dataset.**

|  | Precision | Recall | F1 |
|---|---|---|---|
| Land subsidence | 75.8 | 76.8 | 76.3 |
| Ground fissure | 77.2 | 78.3 | 77.7 |
| Landslide | 80.2 | 78.8 | 79.5 |
| Earthquake | 80.5 | 79.0 | 79.7 |
| Collapse | 76.4 | 77.3 | 76.8 |
| Debris flow | 78.4 | 79.5 | 78.9 |
| Land collapse | 75.5 | 76.7 | 76.1 |

**Table 5   Experimental results of event detection on ACE2005 Chinese dataset.**

|  | TI(%) | | | TC(%) | | |
|---|---|---|---|---|---|---|
|  | Precision | Recall | F1 | Precision | Recall | F1 |
| DMCNN(word) | 66.6 | 63.6 | 65.1 | 61.6 | 58.8 | 60.2 |
| BERT-BILSTM-CRF | 75.6 | 73.1 | 74.3 | 72.7 | 69.4 | 71.0 |
| BERT | 76.5 | 72.2 | 74.2 | 73.3 | 69.0 | 71.1 |
| NPNS | 64.8 | 73.8 | 69.0 | 60.9 | 69.3 | 64.8 |
| C-BILSTM(word) | 68.4 | 61.2 | 64.6 | 63.9 | 57.4 | 60.5 |
| MFEE | 76.9 | 75.4 | 76.2 | 74.3 | 72.9 | 73.6 |
| MFEE(w/o segment) | 77.2 | 75.1 | 76.1 | 74.1 | 72.6 | 73.3 |
| MFEE(w/o BERT) | 76.3 | 73.8 | 75.0 | 73.5 | 71.5 | 72.5 |

sets. From the experimental results, the F1 values of our proposed MFEE for trigger word identification and classification on the ACE 2005 dataset are 76.2% and 73.6%, respectively. There is a performance improvement compared with other models. The generality and effectiveness of the MFEE are verified.

**Ablation Tests.** To verify the effectiveness of our proposed feature enhancement method on the Chinese event extraction task, we conducted a series of comparative experiments on the ACE 2005 dataset. The descriptions of our proposed method and the commonly used lexical introduction methods in previous studies are shown in Table 6. The experimental results are shown in Table 7. Table 7 shows that: (1) Baseline model can achieve 74.3% and 71.0% performance in F1 values for trigger word identification and classification, respectively. Compared with the word-based model, it avoids the mismatch between the word and the triggers word. (2) After introducing the Bi-Char feature representation for the character vector, the F1 values of the model for both trigger word identification and classification decrease. This indicates that Bi-Char does not bring positive semantic information to the Chinese event detection task. (3) Introducing lexical features using a word segmentation tool enhances Chinese event detection. It improved 0.9% and 1.1% on the trigger word identification and classification tasks. Our proposed method of introducing multiple lexical features improves the F1 values by 2.5% and 2.9% for the two tasks, respectively. It indicates that our approach can bring more prosperous and accurate semantic information to the model. (4) We further try to introduce dimensional entity features. It is found that the rest of the metrics of both tasks are improved to some extent, except for a slight decrease of 0.2% in the accuracy rate of trigger word identification. It indicates that introducing entity features can improve the model's performance.

We conducted comparative experiments on lexical feature importance decisions. The methods to determine the lexical weights are as follows: (1) Word average weighted (Avg-weighted): the average of all lexical representations in the lexicon is taken as the feature vector representation. (2) Word frequency weighted (Free-weighted): the lexical feature representation is obtained by a designed lexical feature weighting decision method. (3) Co-determination (Co-determination): the method adopted by MFEE. The lexical weights in the lexicon are co-determined by using a designed lexical feature weighting decision method and word segmentation results. We introduce multi-word lexical features

**Table 6   Contrast model settings.**

| Model | Description |
|---|---|
| Baseline | BERT+BiLSTM+CRF is used as the baseline model. The word feature vector is obtained by the BERT model. Then use BiLSTM to get the sequence context features and constrain the labeling output of the sequence by CRF. |
| Bi-Char | The feature representation of $(c_i, c_{i+1})$ is introduced for word $c_i$ to obtain the semantic interaction information of the preceding and following characters. |
| Segment | The results are obtained using an automatic word segmentation tool. The lexical feature information is introduced concerning the character's relative position. |
| MFEE-entity | All words are introduced for the model in lexical matching subsequences. The importance of lexical features is decided by a designed lexical feature weighting decision method in collaboration with automatic word segmentation tools. |
| MFEE | The entity labeling feature provided by the automatic word segmentation tool is further introduced on top of MFEE-entity. |

**Table 7   Experimental results of feature enhancement method comparison.**

| | TI(%) | | | TC(%) | | |
|---|---|---|---|---|---|---|
| | Precision | Recall | F1 | Precision | Recall | F1 |
| Baseline | 75.6 | 73.1 | 74.3 | 72.7 | 69.4 | 71.0 |
| Bi-Char | 75.6 | 71.6 | 73.6 | 72.1 | 69.0 | 70.4 |
| Segment | 76.4 | 73.6 | 75.2 | 73.3 | 70.9 | 72.1 |
| MFEE-entity | **77.7** | 75.9 | 76.8 | 75.1 | 72.7 | 73.9 |
| MFEE | 77.5 | **76.7** | **77.1** | **75.3** | **73.3** | **74.3** |

for the character representation of the Baseline model (Baseline model setup as described in Table 6) by three different decision methods. The experimental results are shown in Table 8. As can be seen from Table 8, the word averaging weighting method brings limited performance improvement to the ACE 2005 dataset. It indicates that the average weighting method does not enable the model to determine the lexical information that is more consistent with the current contextual semantics. We propose a method to collaboratively designed lexical feature weighting decision method and word segmentation tool. The performance of trigger word detection and trigger word classification tasks is improved. It can be seen that the introduction of multi-word feature information requires the selection of a suitable method.

## CONCLUSION AND FUTURE WORK

We propose a new framework, called MFEE, for Chinese event extraction in the geological hazard domain. It matches multi-word lexical by external lexicon and embeds them into corresponding positions. Using the designed lexical feature weighting decision method and word segmentation tool, the importance of words in the lexicon is decided. The character

**Table 8  Experimental results of the lexical feature weighting decision method.**

| | TI(%) | | | TC(%) | | |
|---|---|---|---|---|---|---|
| | Precision | Recall | F1 | Precision | Recall | F1 |
| Baseline | 75.6 | 73.1 | 74.3 | 72.7 | 69.4 | 71.0 |
| Avg-weighted | 76.8 | 72.6 | 74.6 | 73.4 | 69.1 | 71.2 |
| Fre-weighted | 76.5 | 75.0 | 75.7 | 74.8 | 71.9 | 73.3 |
| Co-determination | 77.7 | 75.9 | 76.8 | 75.1 | 72.7 | 73.9 |

representation is fused with lexical features to enhance semantic capability. We built a large-scale real dataset CGHaz in the geological hazard domain to verify the approach's effectiveness. Experimental results on this dataset and the ACE 2005 dataset demonstrate the effectiveness and generality of MFEE. Our future work may further address the lack of data problem in Chinese event extraction and better extend our framework to the open domain.

### Funding
This work was supported by Science and Technology on Information System Engineering Laboratory [WDZC20205250410] and the Key-Area Research and Development Program of Guangdong Province under Grant [2019B111101001]. The funders had no role in study design, data collection and analysis, decision to publish, or preparation of the manuscript.

### Grant Disclosures
The following grant information was disclosed by the authors:
Science and Technology on Information System Engineering Laboratory: WDZC20205250410.
Key-Area Research and Development Program of Guangdong Province: 2019B111101001.

### Competing Interests
The authors declare there are no competing interests.

### Author Contributions
- Jie Gong conceived and designed the experiments, performed the experiments, analyzed the data, performed the computation work, prepared figures and/or tables, authored or reviewed drafts of the article, and approved the final draft.
- Yang Cao conceived and designed the experiments, performed the experiments, authored or reviewed drafts of the article, and approved the final draft.
- Miao Zijing conceived and designed the experiments, performed the experiments, analyzed the data, performed the computation work, authored or reviewed drafts of the article, and approved the final draft.
- Qiaosen Chen conceived and designed the experiments, performed the experiments, authored or reviewed drafts of the article, and approved the final draft.

## Data Availability

The data is available at Github and Zenodo: https://github.com/JieGong1130/MFEE-dataset; JIE GONG. (2023). JieGong1130/MFEE-dataset: v1.0.0 (v1.0.0). Zenodo. https://doi.org/10.5281/zenodo.7538283.

The code is available at Github and Zenodo: https://github.com/JieGong1130/MFEE-master; JIE GONG. (2023). JieGong1130/MFEE-master: v1.0.0 (v1.0.0). Zenodo. https://doi.org/10.5281/zenodo.7538575.

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
