# Peer review of "MFEE: a multi-word lexical feature enhancement framework for Chinese geological hazard event extraction"

_PeerJ Computer Science, doi:10.7717/peerj-cs.1275_

## Round 0.1 · original submission · Major Revisions

Based on the reviews, the paper has merits, but also has some issues. It can be further considered after improving the paper according to the comments.

Reviewer 1 ·

Basic reporting

The authors propose a Chinese event extraction framework (MFEE) based on character-level sequence labeling that also uses word lexicon information. The approach attempts to guarantee that the lexical information that matches the current semantics is used reasonably. The authors also construct a large Chinese geological hazard dataset (CGHaz). The evaluation is conducted on the new CGHaz and the classical ACE 2005 datasets.

The framework relies on state of the art NLP methods (BiLSTM and BERT) and on classical methods (CRF). Features are obtained at various levels (character, word segmentation, multi-word and entity). A comparative analysis against other architectures and some existing event-extraction proposals demonstrates that the proposed method achieves competitive results.

Experimental design

The authors should discuss in more detail the labeling process and its outcome. It is only mentioned that 673 documents were labeled in total and the ratio of each event type is shown in figure 3 but nothing is said about the average length of the documents and there is no information about the source of the labeled documents. Also, nothing is said about the arguments of each type of event. Did the authors build a taxonomy for the annotation process or did they rely on existing classification codes? What guidelines did the annotators receive?

A total of seven people participated in the labeling process but it is not clear whether the multi-person labeling method was completed by asking the annotators to work collaboratively or it was carried out individually followed by a consensus stage for those labels for which there was no agreement.

Validity of the findings

An analysis of the performance based on each of the seven event types is missing. A detailed analysis and evaluation of the argument extraction performance is also missing. It is not clear whether the reported results refer to named entity recognition and trigger classification or they also include argument extraction (argument identification and role labeling). The reported results should be disaggregated in such a way that it is possible to appreciate the effectiveness of the framework on each of these subtasks.

According to the authors most existing event extraction methods do not specifically target the Chinese geological hazards domain. However, the authors themselves have published the following highly related manuscript and no discussion is provided on the differences between both works:

Yan, J., Chen, L., Yu, Y., Xu, H., Gao, Q., Cao, K., & Chen, J. (2022). EmergEventMine: End-to-End Chinese Emergency Event Extraction Using a Deep Adversarial Network. ISPRS International Journal of Geo-Information, 11(6), 345.

In order to appreciate the novel ingredients of the manuscript that is currently under review, a thorough comparison against existing proposals (including those made by the authors) is required.

Additional comments

Although the authors mention the Chinese geological hazard dataset as an important contribution of their work, the dataset was not made available. Both the code and data should be made available for reproducibility purposes.

The manuscript is well written and easy to follow. I suggest a few changes:

Spaces are missing before “(“ in several parts of the manuscript:

Line 36: feature enhancement framework(MFEE) -> : feature enhancement framework (MFEE)

Starting on line 239: DMCNN(Chen et al.,2015) , C-BiLSTM(Zeng et al.,2016), BERT(Devlin et al., 2019), NPNs(Lin et al.,2018) -> DMCNN (Chen et al.,2015) , C-BiLSTM (Zeng et al.,2016), BERT (Devlin et al., 2019), NPNs (Lin et al.,2018)

Table 3: The heading of the second column should be Description instead of Describe

Reviewer 2 ·

Basic reporting

clear English: yes

intro & background: adequate

structure: clear

figures: Generally fine. Tables (such as Tables 1, 2, ...) might benefit from explaining abbreviations (such as P, R, F1) in the caption.

raw data: Raw data is provided on GitHub. It appears correct to the extent that I could tell.

Experimental design

in scope: yes

questions well defined: yes

rigorous investigation: yes

methods described in detail: Generally yes.

Points that were unclear:

line 134-137 math definition is unclear. For instance, judging by the text in line 131, B(c) seems to include all character sequences starting with c. I don't understand how this maps onto line 134. Why does w have two indices? If c is a character, what does 1 <= c = i mean?

line 229-230: what are lambda1, lambda2. gamma?

Validity of the findings

Conclusions well supported: yes

Additional comments

In general, I found this paper clear and readable. I believe it is in good shape, except for some technical details which are unclear, described under 2.

Minor:

line 209/10: "use" and "are used" is duplicate

Formatting of references should be made consistent.

---

## Round 0.2 · Minor Revisions

The authors have solved the most issues proposed by the reviewers. However, there are some writing issues needing to be revised. I suggest the authors read and check the writing thoroughly.

Reviewer 1 ·

Basic reporting

The authors have addressed the reviewers comments and I'm satisfied with their responses and updates to the manuscript.

Experimental design

No comment

Validity of the findings

No comment

Additional comments

A few typos/grammatical issues need to be fixed:

Line 244 (missing space): labeling process(including -> labeling process (including

Line 296: As Table 2 and Table 3 *shows* -> As Table 2 and Table 31 show

Line 297: Figure 5 and Table 4 *shows* the results of argument extraction for the seven geological hazard event types on the CGHaz dataset.
->
Figure 5 and Table 4 show the results of argument extraction for the seven geological hazard event types on the CGHaz dataset.

Line 362: I suggest rephrasing the first sentence of the conclusion section:
For Chinese event extraction in the geological hazard domain, we propose a new framework, MFEE.
->
We propose a new framework, called MFEE, for Chinese event extraction in the geological hazard domain.

---

## Round 0.3 · accepted · Accept

Thanks for your efforts in improving your manuscript. Congrats!